# Nutritional Status of Children with Cerebral Palsy in Gorkha, Nepal: Findings from the Nepal Cerebral Palsy Register

**DOI:** 10.3390/nu13082537

**Published:** 2021-07-25

**Authors:** Israt Jahan, Mohammad Muhit, Mahmudul Hassan Al Imam, Ratul Ghose, Amir Banjara Chhetri, Nadia Badawi, Gulam Khandaker

**Affiliations:** 1CSF Global, Dhaka 1213, Bangladesh; mmuhit@hotmail.com (M.M.); physiomahmud@yahoo.com (M.H.A.I.); splash348@gmail.com (R.G.); 2Asian Institute of Disability and Development (AIDD), University of South Asia, Dhaka 1212, Bangladesh; 3School of Health, Medical and Applied Sciences, Central Queensland University, Rockhampton, QLD 4701, Australia; gulam.khandaker@health.nsw.gov.au; 4Central Queensland Public Health Unit, Central Queensland Hospital and Health Service, Rockhampton, QLD 4700, Australia; 5CSF Global-Nepal, Balaju, Kathmandu 44611, Nepal; amirbanjara@yahoo.com; 6Cerebral Palsy Alliance Research Institute, The University of Sydney, Sydney, NSW 2086, Australia; nadia.badawi@health.nsw.gov.au; 7Grace Centre for Newborn Intensive Care, Sydney Children’s Hospital Network, Westmead, NSW 2145, Australia; 8Discipline of Child and Adolescent Health, Sydney Medical School, The University of Sydney, Sydney, NSW 2006, Australia

**Keywords:** cerebral palsy (CP), malnutrition, disability, key informant method, Nepal

## Abstract

Background: The study aimed to define the burden and underlying risk factors of malnutrition among children with cerebral palsy (CP) in Gorkha district, Nepal. Methods: The first population-based register of children with CP in Gorkha, Nepal (i.e., Nepal CP Register—NCPR) was established in 2018. Children aged <18 years with confirmed CP were registered following standard protocol. Nutritional status was determined based on anthropometric measurements (height/length, weight, mid-upper-arm-circumference) following WHO guidelines. Descriptive analyses and adjusted logistic regression were completed. Results: Between June–October 2018, 182 children with CP were registered into the NCPR (mean (SD) age at assessment: 10.3 (5.0) years, 37.4% female). Overall, 51.7%, 64.1%, and 29.3% children were underweight, stunted, and thin, respectively. Furthermore, 14.3% of children with CP aged <5 years had severe wasting. Underweight and stunting were significantly higher among children with spastic CP (*p* = 0.02, *p* < 0.001) and/or Gross Motor Function Classification System (GMFCS) level (III–V) (*p* = 0.01, *p* < 0.001) and/or who were not enrolled in school (*p* = 0.01, *p* < 0.001). In adjusted analysis, GMFCS level III–V and non-attendance to school significantly increased the odds of stunting by 8.2 (95% CI 1.6, 40.8) and 4.0 (95% CI 1.2, 13.2) times, respectively. Conclusions: the high rate of different forms of undernutrition among children with CP in Gorkha, Nepal is concerning. Need-based intervention should be taken as priority to improve their nutritional outcome.

## 1. Introduction

Cerebral Palsy (CP) is a non-progressive neurological condition that results from lesion to the developing brain [1]. The impact of CP on a child’s life is multidimensional and not limited to functional impairment. The motor impairment also affects nutritional status, quality of life, oral health and even survival probability of a child with CP [2,3,4,5,6]. Maintenance of adequate nutrition is critical and malnutrition is common among this vulnerable population [2,3,4,7,8,9]. Although in recent years increased number of studies have focused on the nutritional status of children with CP [2,3,4,8,9,10,11,12], the evidence gap in low- and middle-income countries (LMICs) is still high.

In a recent population-based study in Bangladesh, the prevalence of CP was estimated at 3.4 per 1000 live births which is nearly twice that estimated globally [13,14]. In the same cohort, 70% of children were underweight and/or stunted [2]. A similar high burden of malnutrition was also reported from Indonesia and Uganda [3,9]. The causal pathway to malnutrition among children with CP is yet not clearly defined particularly in LMIC settings, though several studies found a significant association between malnutrition, severity of motor impairment (e.g., Gross Motor Function Classification System (GMFCS) level III–V, tri/quadriplegia) and associated impairments (e.g., intellectual, speech, hearing) among children with CP [2,3,4,7,15]

In Nepal, very little is known about the epidemiology of CP. The limited available institution-based studies indicate a substantial higher burden of low economic status, delayed diagnosis, and severe motor impairment among children with CP in the country [16,17,18]. However, none of the available studies reported the nutritional status of children with CP in Nepal. This gap in evidence on the epidemiology of malnutrition (e.g., burden, preventable and manageable risk factors) remains a major obstacle for planning targeted nutritional intervention for children with CP in a geographically complex LMIC such as Nepal. Population-based studies hold great potential to contribute reliable evidence and serve as baseline data for such intervention studies as well as program planning. This study aimed to describe the burden and underlying risk factors of malnutrition among children with CP using population-based data in remote Gorkha district of Nepal.

## 2. Materials and Methods

### 2.1. Participants and Settings

This study used data from the first population-based register of children with CP (<18 years) in six municipalities of remote Gorkha district, Nepal, i.e., Nepal CP Register—NCPR. The NCPR was established in 2018 and the study area encompasses ~823sq km area and 46,681 households with a total population ~184,546 (child population aged <18 years~83,047) [19,20]

The community-based key informant method (KIM, a validated and widely used method) [3,13,21] was used to identify children with suspected CP. A total of 174 key informants (KIs—local volunteers) were trained as part of NCPR. Following training, the KIs received six weeks to identify children with suspected CP and share their name and contact information with the community mobilizers (CM, i.e., paid project staff). The CMs then, with help from the KIs, brought all these children and their primary caregivers to medical camps. Each child underwent detailed neurodevelopmental assessment by a multidisciplinary medical team including a pediatrician, a nutritionist, and a physiotherapist for confirmed clinical diagnosis, registration into the NCPR and data collection [13]. The clinical definition of CP used in NCPR was adopted from the Bangladesh Cerebral Palsy Register (BCPR), [13] based on the Surveillance of Cerebral Palsy in Europe (SCPE) [22] and the Australian Cerebral Palsy Register (ACPR) [23].

### 2.2. Measures

Detailed information about socio-demographic characteristics, pre- and perinatal history, motor function severity, associated impairments, anthropometric measurements, and educational status of the children registered into the NCPR (i.e., children with confirmed CP) were collected. Structured guidelines and a data collection template adopted from BCPR and ACPR were used [13,23].

### 2.3. Clinical Assessment

GMFCS and Manual Ability Classification System (MACS) were used to assess and define motor function severity following standard guidelines [24,25]. The Viking Speech Scale (VSS) and Communication Function Classification System (CFCS) were used to describe the motor speech disorder and functional communication of registered children [26,27]. Presence and severity of associated impairments were determined based on clinical examination, review of available medical records and detailed history provided by the primary caregivers. All assessments were completed following the BCPR protocol [13].

### 2.4. Anthropometric Measurements

Weight: Weight was measured in kilograms using a digital weighing scale with a precision of 10 g following the WHO protocol [28]; three repeated measures were taken, and the average was documented. Tared weight was measured for young children aged <2 years and children who had difficulties in standing independently.

Length/height: Recumbent length/height was measured in centimeters (cm). Recumbent length was measured for children aged <2 years using a manual length board with a precision of 1 cm. For children aged ≥2 years, height was measured using a height board following WHO protocol [28]. Segmental measurement i.e., knee height, was measured for children who could not stand independently due to muscle stiffness, weakness or deformities. The full body length/height was estimated using following formula; height = (2.69 × knee height) + 24.2 in cm [29].

Mid-upper arm circumference (MUAC): MUAC was measured in cm using MUAC tape following the WHO protocol [28].

### 2.5. Indicators Used to Define the Nutritional Status of Children

The following indicators were used to measure the nutritional status of children registered in the NCPR, (i) weight-for-age z score (WAZ), (ii) height-for-age z score (HAZ), (iii) weight-for-height z score (WHZ), (iv) BMI-for-age z score (BAZ), and (v) MUAC-for-age z score (MUACZ). All z scores were calculated using WHO Anthro and WHO AnthroPlus software. HAZ and BAZ were calculated for children aged <18 years, WAZ was calculated for children aged <10.1 years, and WHZ and MUACZ were calculated for children aged <5.1 years. The z scores were categorized using the WHO cut-off values (overnutrition: z score > +2 SD, normal: z score −2.0 to +2.0 SD, moderate undernutrition: z score ≥ −3.0 to <−2.0 SD, severe undernutrition: z score < −3.0 SD) to determine the nutritional status of children [28]. A child was considered underweight if WAZ < −2.0 SD, stunted if HAZ < −2.0 SD, wasted if WHZ or MUACZ < −2.0 SD, and thin if BAZ < −2.0 SD. Venn diagram was used to show the overlapping of different forms of undernutrition among the participating children.

### 2.6. Statistical Analysis

Data were analyzed using SPSS (IBM Corporation, Chicago, IL, USA) version 26. Skewness and Kurtosis were used to examine the distribution of continues variables, e.g., age, monthly family income, z scores. Descriptive analyses were completed using valid percentages to report the overall nutritional status and bivariate analyses were used to identify potential determinants of undernutrition among children in NCPR. Chi-squared test and Fisher’s exact test were used. Factors that were found significantly related to underweight and/or stunting and/or thinness among participating children in the cross tabulation (e.g., chi-square, fisher’s exact test) were included in unadjusted logistic model as potential predictors. Factors significantly associated with underweight or stunted or thinness in unadjusted analyses were entered into the adjusted model i.e., logistic regression; adjusted odds ratios (aOR) with 95% confidence intervals (95% CIs) were reported. A *p* < 0.05 was considered significant.

## 3. Results

Between June and October 2018, 213 children with suspected CP underwent detailed neurodevelopmental assessment in 13 medical assessment camps. Of these, 182 had a clinically confirmed diagnosis of CP. The mean (SD) age at assessment was 10.3 (5.0) years, 37.4% (*n* = 68) female. Of all, 45.1% (*n* = 82/182) were living in the rural areas.

### 3.1. Overall Nutritional Status

Over two thirds (78.2%, *n* = 136/174, missing data, *n* = 8) of the children had at least one form of malnutrition. Of all, 51.7% (*n* = 45/87) were underweight (i.e., WAZ < −2 SD), 64.1% (*n* = 109/170) were stunted (i.e., HAZ < −2 SD) and 29.3% (*n* = 48/164) were thin (i.e., BAZ < −2 SD). Among children aged ≤5 years, 14.3% (*n* = 4/28) had severe wasting i.e., severe acute malnutrition (SAM) according to MUACZ (i.e., MUACZ < −3SD); however, this proportion was 7.7% (*n* = 2/26) when WHZ was used as an indicator (i.e., WHZ < −3SD). The mean (SD) and median [inter quartile range (IQR)] of the z scores have been summarized in Table 1.

### 3.2. Socio-Demographic Characteristics and Nutritional Status of Children with CP

Table 2 summarizes the relationship between socio-demographic characteristics and nutritional status of children with CP in the study area. No significant difference in underweight and stunting was observed among children with CP with changes in educational level of father (*p* = 0.58 and *p* = 0.87, respectively), and mother (*p* = 0.06 and *p* = 0.05, respectively), occupation of father (*p* = 0.38 and *p* = 0.48, respectively) and mother (*p* = 0.70 and *p* = 0.29, respectively), source of drinking water (*p* = 0.72 and *p* = 0.76, respectively) and access to sanitation (*p* = 0.60 and *p* = 0.33, respectively). Similar observations were made for thinness among children. However, underweight was more prevalent among children with low monthly family income compared to others in the cohort (59.1% if <150USD vs. 13.6% if ≥300 USD; *p* = 0.03). A similar pattern was observed for stunting and thinness, but this relationship was not statistically significant (*p* = 0.09 and *p* = 0.93 respectively). Furthermore, school enrolment was significantly lower among school-aged children who were underweight (*p* = 0.01), stunted (*p* < 0.001) and thin (*p* = 0.02) in the NCPR cohort.

### 3.3. Clinical Characteristics and Nutritional Status of Children with CP

#### 3.3.1. Birthweight and Gestational Age

All children with CP in NCPR who had low birth weight also showed stunting (*p* < 0.001). However, nutritional status of children did not vary with their gestational age. (Table 3)

#### 3.3.2. Timing of CP and Age of CP Diagnosis

There was no significant difference in WAZ, HAZ and BAZ according to the timing of CP (*p* = 0.17, *p* = 0.12, *p* = 0.78, respectively) of children. (Table 3)

Although underweight and thinness were observed slightly higher among children who were diagnosed with CP before 30 months of age than those diagnosed after 60 months of age, the differences were not statistically significant (46.2% vs. 28.2%, *p* = 0.85 and 50.0% vs. 32.6%, *p* = 0.54, respectively). Stunting, on the other hand, were significantly overrepresented among children diagnosed with CP aged 0–30 months than others in the cohort (*p* = 0.02). (Table 3)

#### 3.3.3. Predominant Type, Topography, Motor Function Severity, Motor Speech Disorder and Functional Communication

Underweight and stunting were more prevalent among children with spastic CP (*p* = 0.02 and *p* < 0.001, respectively). Moreover, stunting was overrepresented among children with tri/quadriplegia and children with VSS level III–V (*p* = 0.002 and *p* = 0.02 respectively). Furthermore, stunting and thinness were comparatively higher among children with MACS level III–V (*p* = 0.002 and *p* = 0.02 respectively) and CFCS level III–V (*p* = 0.02 and *p* = 0.03 respectively). All three forms of undernutrition (i.e., underweight, stunting, thinness) were significantly higher among children with GMFCS level III–V (*p* = 0.01, *p* < 0.001, *p* = 0.01 respectively). (Table 3)

A significant positive relationship between the ‘GMFCS level and nutritional status’ and ‘nutritional status and school attendance’ was also observed among the participating children. Both school attendance and nutritional status was considerably low among children with GMFCS level III–V than children with GMFCS level I–II (Table 2 and Table 3, and Appendix A).

#### 3.3.4. Associated Impairments

Underweight, stunting and thinness were overrepresented among children with at least one associated impairment. Furthermore, stunting was more prevalent among children with intellectual impairment (*p* = 0.01). However, no significant relationship between nutritional status and other associated impairments (e.g., epilepsy, visual, hearing, speech impairment) were observed in the cohort (*p* ≥ 0.05 for all). (Table 3)

### 3.4. Predictors of Underweight, Stuntin and Thinness Among Children with CP

#### 3.4.1. Underweight

In unadjusted analysis, sex, attendance to mainstream school and GMFCS level were found to be significantly associated with underweight. When fitted in an adjusted model, none of those factors remained significant (*p* = 0.30, *p* = 0.08, *p* = 0.16, respectively). (Appendix A and Table 4)

#### 3.4.2. Stunting

In case of stunting, sex, attendance to mainstream school, GMFCS level, MACS level, VSS level, CFCS level and intellectual impairment were identified as significant predictors. When adjusted for other covariates, attendance to mainstream school, GMFCS level and MACS level of child were found to be significantly associated with stunting. Children who never attended mainstream school and children with GMFCS level III–V had 4.0 (95% CI 1.2, 13.2) and 8.2 (95% CI 1.6, 40.8) times higher odds of stunting when compared to others in the cohort, respectively. (Appendix A and Table 4)

#### 3.4.3. Thinness

In unadjusted analysis, the odds of being thin was significantly higher among children with GMFCS level III–V, MACS level III–V, CFCS level III–V, and those who were not enrolled in mainstream school. When fitted in adjusted model, none of those predictors remained significant. (Appendix A and Table 4)

### 3.5. Children with Multiple Forms of Undernutrition

In the NCPR cohort, 12 children had underweight, stunting and thinness. The majority had the most severe forms of motor impairment (e.g., GMFCS level III–V: *n* = 10/12; tri/quadriplegia: *n* = 9/12), severe speech motor disorder (VSS level III–V: *n* = 10/12) and low functional communication level (CFCS level III–V: *n* = 9/12), associated impairments (e.g., multiple associated impairments: *n* = 4/12). Furthermore, all were from ultra-poor families and half (*n* = 6/12) had never received any rehabilitation services. (Figure 1)

## 4. Discussion

To our knowledge, this is the first population-based study reporting nutritional status of children with CP in Nepal. The study revealed a high burden of malnutrition among children with CP in remote Gorkha, Nepal. Nearly two-thirds of children in NCPR had at least one form of malnutrition, and more than one-third had severe undernutrition. When compared to regional data, a considerably higher rate of wasting was observed among children with CP aged <5 years than the general population of similar age [30]. A similar high burden was reported in other LMICs [2,3,4,9,31].

Malnutrition among children with CP is complex and is due to multiple interlinked factors. It is evident that children with severe motor impairments are the most vulnerable. Several studies have reported a strong relationship between the GMFCS level and malnutrition among children with CP in both LMICs and HICs [2,4,15,32]. Similar findings have been observed in this study. There is a two way relationship between motor severity and malnutrition among children with CP. Children with severe CP (e.g., GMFCS level III–V) are likely to have severe feeding difficulties, poor digestive capacity and altered nutritional requirement [31,32,33,34,35,36,37]. These factors directly affect the nutrient intake, and thus their nutritional status. On the other hand, malnutrition impairs the cerebral function of children, weakens their immune system to fight against infections, and interferes in nutritional intake/absorption causing growth failure [37]. Early identification of malnutrition and comprehensive nutritional support is necessary to break this perpetual relationship and avert the adverse consequences of malnutrition among children with CP. Considering the complex geography of the study site, it is likely that most children in NCPR did not have access to health care and rehabilitation services as needed.

Moreover, children who had speech impairment and severe communication limitations had poor nutritional status than others in the cohort. Communication plays a vital role in maintaining daily activities including feeding management of children with CP. Although not significant, the nutritional status of children with intellectual disabilities was comparatively poor than others in NCPR. Advancement in assistive technologies and evidence-based guidelines developed in recent decades have enhanced opportunities for learning and active participation of children with CP in daily activities, thus improving quality of life. However, such facilities are rarely accessible to children from low socio-economic status and remote settings like Gorkha. The findings suggest that children who were attending schools had a slightly better nutritional status than those not attending. This could be explained by low school attendance and higher malnutrition rates observed among children with severe GMFCS level in the NCPR cohort.

A high proportion of stunting and underweight was observed among children who had LBW and/or were born preterm. Similar positive correlation was reported previously among children with CP in Bangladesh and Indonesia [2,3]. LBW is considered as one of the major predisposing factors for chronic undernutrition, i.e., stunting among children. The recent Nepal Demographic and Health Survey (NDHS) also indicates a high proportion of malnutrition among children who had very small size at birth [30].

Nutrition is crucial and influences every aspect of life to different extents. Because of the long-term impact on health, quality of life and survival, it is important to address malnutrition at an early stage of life and break the vicious cycle. With the growing evidence regarding vulnerability of children with CP, it is now apparent that there is substantial need for action and implementation of evidence-based programming to improve their nutritional status. Up to now, several interventions have been proposed to improve the nutritional outcome of children with CP in HICs and LMICs [37,38,39]. However, in LMICs strategies mostly focused on behavioral interventions, e.g., training to the caregivers on feeding skills, nutrition education [38,39], and the impact of such interventions on nutritional outcome of children with CP in LMICs like Nepal is unknown. Community-based nutrition intervention/programs focusing on need and nutritional outcome could be beneficial.

In recent decades Nepal has showed remarkable strides in reducing the burden of chronic malnutrition among children nationally [30]. Combating childhood malnutrition is one of the key priorities of the government of Nepal, and several initiatives have been undertaken to improve the current scenario. Nevertheless, it is not clear where children with disability, e.g., CP, stand in this agenda. Although the government of Nepal has initiated a social security scheme for people with disability i.e., disability card and pension, several challenges in accessing these services have been reported [40]. The key challenge remains covering the evidence gap, coordination, and collaboration in translational research for children with disability in Nepal.

### Strengths and Limitations

The main strength of this study is that it used population-based data to report the burden and the underlying factors of malnutrition among children with CP in the remote Gorkha district of Nepal. The use of multiple indicators assisted in data triangulation, identification of children suffering from acute and chronic malnutrition, and potential predictors of malnutrition, using adjusted analysis. However, despite considerable effort, this study has several limitations. First, weight, height and MUAC were used to assess nutritional status, which have limitations in the precise estimation of the nutritional status of children with CP; therefore, triceps skin-fold thickness and arm fat area are considered as recommended measurements to assess this nutritional status [37]. However, considering the resource constraints (i.e., trained professional, standard skinfold calipers) and complex study settings, these were the optimal choice to minimize error and ensure the quality of the collected data. Second, the anthropometric data were compared to WHO reference curves developed for the general population as a CP specific growth chart is not available for children in LMICs. Third, the causal pathway to malnutrition among children with CP in Gorkha, Nepal, could not be established as it was beyond the scope of this cross-sectional study design. Fourth, the study sites (i.e., Gorkha) represent the hilly regions of the country, therefore the findings reported in this study are not generalizable to the other parts of the country (e.g., plain land/Tarai). However, this study described the epidemiology of malnutrition among children with CP in Gorkha, Nepal, and these findings will help plan future research and nutrition interventions in such complex settings as that of Nepal.

## 5. Conclusions

The burden of malnutrition is high among children with CP in Gorkha, Nepal. Although Nepal has demonstrated progress in addressing the challenges toward improving nutritional status of children, disability specific data are lacking. The study findings have identified potential predictors of malnutrition, essential for development of need-based nutrition intervention strategies for this vulnerable group of population. The findings are important for program planners and policy makers working in the disability sector in Nepal to prioritize inclusion of these vulnerable children, enhancing opportunity for their development and promotion of a disability inclusive society in Nepal.

## Figures and Tables

**Figure 1 nutrients-13-02537-f001:**
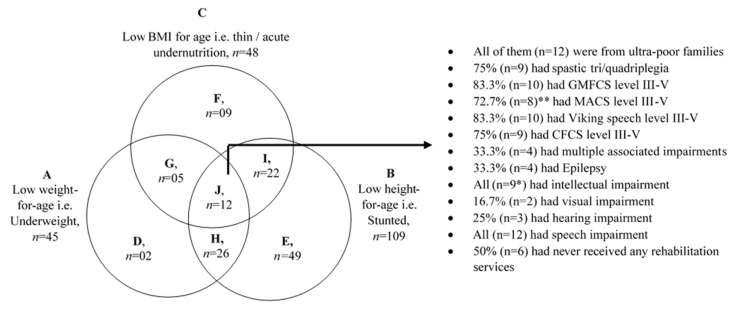
Overlapping of different forms of undernutrition among children with CP in NCPR. Here, (A) = Total number of children who were underweight with or without other forms of undernutrition; (B) = total number of children who were stunted with or without other forms of undernutrition; (C) = total number of children who were thin with or without other forms of undernutrition; (D) = children who were underweight only; (E) = children who were stunted only; (F) = children who were thin only; (G) = children who were both underweight and thin; (H) = children who were both underweight and stunted; (I) = children who were both stunted and thin; (J) children who had all three forms of undernutrition (i.e., underweight, stunting, thinness); * Missing data; ** was assessed for children aged at or over 4 years.

**Table 1 nutrients-13-02537-t001:** Overall nutritional status of children registered into the Nepal CP Register (NCPR).

Indicator	Weight-for-Age z Score (WAZ)	Height-for-Age z Score (HAZ)	BMI-for-Age z Score (BAZ)	Weight-for-Height-z Score (WHZ)	MUAC-for-Age-z Score (MUACZ)
*n*	87 ^1^	170 ^2^	164 ^3^	26 ^2,4^	28 ^2,4^
Mean (SD)	−2.2 (1.9)	−2.9 (2.6)	−0.5 (4.1)	−0.5 (1.6)	−0.9 (1.4)
Median (IQR)	−2.1 (−3.8, −0.9)	−2.8 (−4.5, −1.4)	−1.1 (−2.5, 0.6)	−0.1 (−1.7, 0.5)	−0.6 (−2.2, 0.1)
Overnutrition, *n* (%) (z score: >+2 SD)	4 (4.6)	6 (3.5)	13 (7.9)	1 (3.8)	0 (0.0)
Normal, *n* (%) (z score: −2 SD to +2 SD)	38 (43.7)	55 (32.4)	103 (62.8)	21 (80.8)	21 (75.0)
Moderate undernutrition, *n* (%) (z score: >−3 SD to <−2.0 SD)	11 (12.6)	30 (17.6)	19 (11.6)	2 (7.7)	3 (10.7)
Severe undernutrition, *n* (%) (z score: ≤−3.0 SD)	34 (39.1)	79 (46.5)	29 (17.7)	2 (7.7)	4 (14.3)

^1^ Weight for age was calculated for children aged 0–121 months; ^2^ Missing data; ^3^ Weight for height was calculated for children aged 0–61 months; ^4^ MUAC for age z score was calculated for children aged 0–61 months.

**Table 2 nutrients-13-02537-t002:** Weight-for-age, height-for-age and BMI-for-age of children registered into NCPR according to their socio-demographic characteristics.

Characteristics	Total *n* = 182	Weight for Age z Score (WAZ), *n* = 83 ^1^		*p* Value ^4^	Height for Age z Score (HAZ), *n* = 164 ^2^		*p* Value ^4^	BMI for Age z Score (BAZ), *n* = 151 ^3^		*p* Value ^4^
		Normal, *n* = 38	Underweight, *n* = 45		Normal, *n* = 55	Stunted, *n* = 109		Normal, *n* = 103	Thin, *n* = 48	
**Age group (in years), *n* = 181 ^6^**										
0–4	26 (14.4)	15 (40.5)	9 (20.4)	0.08 ^5^	9 (16.4)	12 (11.1)	0.51	17 (16.7)	5 (10.4)	0.51
5–9	63 (34.8)	22 (59.5)	34 (77.3)		21 (38.2)	37 (34.3)		36 (35.3)	16 (33.3)	
10–14	54 (29.8)	0	1 (2.3) ^2^		13 (23.6)	37 (34.3)		31 (30.4)	14 (29.2)	
≥15	38 (21.0)	-	-		12 (21.8)	22 (20.4)		18 (17.6)	13 (27.1)	
**Sex, *n* = 182**										
Male	114 (62.6)	28 (73.7)	23 (51.1)	**0.03**	41 (74.5)	62 (56.9)	**0.03**	65 (63.1)	29 (60.4)	0.75
Female	68 (37.4)	10 (26.3)	22 (48.9)		14 (25.5)	47 (43.1)		38 (36.9)	19 (39.6)	
**Mother’s educational level (completed years of formal schooling) ^6^, *n* = 170**										
0–4	113 (66.5)	14 (42.4)	28 (63.6)	0.06	28 (56.0)	77 (74.8)	0.05	64 (66.7)	33 (68.8)	0.44
5–9	32 (18.8)	7 (21.2)	10 (22.7)		11 (22.0)	15 (14.6)		15 (15.6)	10 (20.8)	
≥10	25 (14.7)	12 (36.4)	6 (13.6)		11 (22.0)	11 (10.7)		17 (17.7)	5 (10.4)	
**Father’s educational level (completed years of formal schooling)^6^, *n* = 167**										
0–4	77 (46.1)	13 (37.1)	21 (47.7)	0.58	25 (50.0)	46 (46.0)	0. 87	45 (47.9)	19 (41.3)	0.62
5–9	52 (31.1)	11 (31.4)	13 (29.5)		16 (32.0)	33 (33.0)		29 (30.9)	18 (39.1)	
≥10	38 (22.8)	11 (31.4)	10 (22.7)		9 (18.0)	21 (21.0)		20 (21.3)	9 (19.6)	
**Mother’s occupation at birth of child with CP ^6^, *n* = 167**										
Agricultural work	81 (48.5)	13 (37.1)	20 (47.6)	0.70 ^5^	20 (40.0)	51 (51.5)	0.29 ^5^	42 (43.8)	25 (56.8)	0.46 ^5^
Business	11 (6.6)	3 (8.6)	4 (9.5)		2 (4.0)	7 (7.1)		6 (6.3)	2 (4.5)	
No IGA	66 (39.5)	16 (45.7)	14 (33.3)		26 (52.0)	35 (35.4)		41 (42.7)	16 (36.4)	
Other IGA	9 (5.4)	3 (8.6)	4 (9.5)		2 (4.0)	6 (6.1)		7 (7.3)	1 (2.3)	
**Father’s occupation at birth of child with CP ^6^, *n* = 171**										
Agricultural work	59 (34.5)	10 (27.8)	15 (34.9)	0.38 ^5^	17 (33.3)	36 (35.3)	0.48 ^5^	33 (34.0)	12 (26.1)	0.62 ^5^
Business	11 (6.4)	3 (8.3)	2 (4.7)		1 (2.0)	8 (7.8)		5 (5.2)	2 (4.3)	
No IGA	8 (4.7)	0 (0.0)	3 (7.0)		3 (5.9)	4 (3.9)		5 (5.2)	1 (2.2)	
Other IGA	93 (54.4)	23 (63.9)	23 (53.5)		30 (58.8)	54 (52.9)		54 (55.7)	31 (67.4)	
**Monthly family income, NRs (USD) ^6,7^, *n* = 181**										
Median (IQR)	10,000 (7000, 20,000) (82 (57, 164))	17,000 (9750, 20,500) (140 (80.0 168))	12,000 (7000, 20,000) (98 (57, 164))	0.14	12,000 (8000, 20,000)(98 (66, 164))	10,500 (7000, 20,000) (86 (57, 164))	0.88	12,000 (7000, 20,000) (98 (57, 164)	13,500 (8250, 20,000) (111 (68, 164))	0.55
1000–14,999 (10–149.9)	103 (56.9)	13 (34.2)	26 (59.1)	**0.03** ^5^	29 (52.7)	62 (57.4)	0.09	54 (52.9)	24 (50.0)	0.93
15,000–29,999 (150–299.9)	61 (33.7)	21 (55.3)	12 (27.3)		24 (43.6)	33 (30.6)		37 (36.3)	19 (39.6)	
≥30,000 (≥300)	17 (9.4)	4 (10.5)	6 (13.6)		2 (3.6)	13 (12.0)		11 (10.8)	5 (10.4)	
**Source of drinking water ^6^, *n* = 177**										
Improved ^8^	163 (92.6)	31 (86.1)	40 (90.9)	0.72 ^5^	48 (90.6)	99 (92.5)	0.76 ^5^	91 (91.9)	43 (89.6)	0.76 ^5^
Unimproved	13 (7.4)	5 (13.9)	4 (9.1)		5 (9.4)	8 (7.5)		8 (8.1)	5 (10.4)	
Type of latrin used ^6^										
Sanitary ^9^	169 (97.1)	34 (94.4)	40 (97.6)	0.60 ^5^	48 (94.2)	103 (98.1)	0.33 ^5^	95 (96.9)	44 (95.7)	0.65 ^5^
Non-sanitary	5 (2.9)	2 (5.6)	1 (2.4)		3 (5.8)	2 (1.9)		3 (3.1)	2 (4.3)	
**Attendance at mainstream and/or special school ^6^, *n* = 174**										
No	83 (47.7)	7 (18.4)	20 (47.6)	**0.01**	14 (26.4)	63 (60.0)	**<0.001**	37 (37.4)	28 (59.6)	**0.02**
Yes	53 (30.5)	13 (34.2)	6 (14.3)		28 (52.8)	22 (21.0)		39 (39.4)	9 (19.1)	
N/A (aged <6 year)	38 (21.8)	18 (47.4)	16 (38.1)		11 (20.8)	20 (19.0)		23 (23.2)	10 (21.3)	

^1^ WAZ was calculated for children aged 0–121 months and, due to very small counts, the overweight children (WAZ > +2 SD) were excluded from analysis (*n* = 4); ^2^ HAZ > +2 SD were excluded from analysis due to very small counts (*n* = 6); ^3^ BAZ > +2 SD were excluded from analysis due to very small counts (*n* = 13); ^4^ Chi-square test (two-tailed); ^5^ Fisher’s exact test (two-tailed); ^6^ Missing data; ^7^ 1 USD~121 NRs; ^8^ Improved drinking water source was defined following Nepal Demographic and Health Survey (NDHS) categories which included piped water/tube well/protected well/bottled water; ^9^ Sanitary latrines included flush toilet/pit latrine. The *p* values in ‘**bold**’ font indicate significant difference.

**Table 3 nutrients-13-02537-t003:** Weight-for-age, height-for-age and BMI-for-age of children registered into the NCPR according to their clinical characteristics.

Risk Factors and Clinical Characteristics	Total, *n* = 182	Weight for Age z Score (WAZ) ^1^, *n* = 83		*p* Value ^4^	Height for Age z Score (HAZ) ^2^, *n* = 164		*p* Value ^4^	BMI for Age z Score (BAZ) ^3^, *n* = 151		*p* Value ^4^
		Normal, *n* = 38	Underweight, *n* = 45		Normal, *n* = 55	Stunted, *n* = 109		Normal, *n* = 103	Thin, *n* = 48	
**Birthweight (BW) among known ^6^, *n* = 78**										
Normal BW	57 (73.1)	19 (79.2)	19 (57.6)	0.09	22 (100.0)	29 (60.4)	**0.001**	35 (74.5)	14 (66.7)	0.51
Low BW	21 (26.9)	5 (20.8)	14 (42.4)		0 (0.0)	19 (39.6)		12 (25.5)	7 (33.3)	
**Gestational age ^6^, *n* = 170**										
Pre-term	18 (10.6)	4 (12.1)	8 (18.6)	0.44	3 (6.1)	15 (14.6)	0.13	14 (14.7)	3 (6.5)	0.16
Term	152 (89.4)	29 (87.9)	35 (81.4)		46 (93.9)	88 (85.4)		81 (85.3)	43 (93.5)	
**Epilepsy, *n* = 182**										
No	132 (72.5)	24 (63.2)	33 (73.3)	0.32	41 (74.5)	75 (68.8)	0.45	78 (75.7)	31 (64.6)	0.15
Yes	50 (27.5)	14 (36.8)	12 (26.7)		14 (25.5)	34 (31.2)		25 (24.3)	17 (35.4)	
**Intellectual ^6^, *n* = 132**										
No	47 (35.6)	14 (50.0)	9 (26.5)	0.06	21 (51.2)	23 (28.7)	**0.01**	35 (46.1)	9 (27.3)	0.07
Yes	85 (64.4)	14 (50.0)	25 (73.5)		20 (48.8)	57 (71.3)		41 (53.9)	24 (72.7)	
**Visual ^6^, *n* = 179**										
No	161 (89.9)	32 (84.2)	37 (84.1)	1.00	49 (92.5)	95 (88.0)	0.38	91 (90.1)	41 (87.2)	0.60
Yes	18 (10.1)	6 (15.8)	7 (15.9)		4 (7.5)	13 (12.0)		10 (9.9)	6 (12.8)	
**Hearing ^6^, *n* = 179**										
No	143 (79.9)	29 (76.3)	33 (73.3)	0.76	42 (79.2)	86 (78.9)	0.96	83 (81.4)	36 (78.3)	0.66
Yes	36 (20.1)	9 (23.7)	12 (26.7)		11 (20.8)	23 (21.1)		19 (18.6)	10 (21.7)	
**Speech ^6^, *n* = 181**										
No	36 (19.9)	10 (26.3)	6 (13.3)	0.13	16 (29.6)	18 (16.5)	0.05	26 (25.5)	6 (12.5)	0.07
Yes	145 (80.1)	28 (73.7)	39 (86.7)		38 (70.4)	91 (83.5)		76 (74.5)	42 (87.5)	
**Number of associated impairments, *n* = 182**										
None	23 (12.6)	6 (15.8)	5 (11.1)	0.80	12 (21.8)	9 (8.3)	**0.03**	17 (16.5)	5 (10.4)	0.28
≤2	113 (62.1)	20 (52.6)	26 (57.8)		33 (60.0)	68 (62.4)		65 (63.1)	28 (58.3)	
Multiple	46 (25.3)	12 (31.6)	14 (31.1)		10 (18.2)	32 (29.4)		21 (20.4)	15 (31.3)	
**Timing of CP ^6^, *n* = 180**										
Pre & Peri	159 (88.3)	31 (81.6)	42 (93.3)	0.17 ^5^	45 (83.3)	99 (91.7)	0.12	88 (88.1)	43 (89.6)	0.78
Postnatal	21 (11.7)	7 (18.4)	3 (6.7)		9 (16.7)	9 (8.3)		12 (11.9)	5 (10.4)	
**Swallowing difficulties ^6^, *n* = 179**										
No	117 (65.4)	24 (63.2)	24 (54.5)	0.43	40 (74.1)	67 (61.5)	0.11	69 (67.6)	29 (61.7)	0.48
Yes	62 (34.6)	14 (36.8)	20 (45.5)		14 (25.9)	42 (38.5)		33 (32.4)	18 (38.3)	
**Predominant type of CP, *n* = 182**										
Spastic	141 (77.5)	24 (63.2)	39 (86.7)	**0.02** ^5^	39 (70.9)	89 (81.7)	**<0.001** ^5^	74 (71.8)	40 (83.3)	0.42 ^5^
Dyskinesia	7 (3.8)	2 (5.3)	0 (0.0)		2 (3.6)	4 (3.7)		4 (3.9)	2 (4.2)	
Ataxia	16 (8.8)	7 (18.4)	1 (2.2)		12 (21.8)	2 (1.8)		11 (10.7)	3 (6.3)	
Hypotonia	18 (9.9)	5 (13.2)	5 (11.1)		2 (3.6)	14 (12.8)		14 (13.6)	3 (6.3)	
**Topography of CP, *n* = 141**										
Mono/Hemiplegia	54 (38.3)	10 (41.7)	11 (28.2)	**0.54** ^5^	24 (61.5)	26 (29.2)	**0.002** ^5^	31 (41.9)	12 (30.0)	0.18 ^5^
Diplegia	16 (11.3)	3 (12.5)	5 (12.8)		4 (10.3)	10 (11.2)		10 (13.5)	3 (7.5)	
Tri/Quadriplegia	71 (50.4)	11 (45.8)	23 (59.0)		11 (28.2)	53 (59.6)		33 (44.6)	25 (62.5)	
**CP diagnosis age (in months) ^6^, *n* = 168**										
≤30	78 (46.4)	19 (52.8)	18 (46.2)	0.85	21 (42.9)	48 (46.2)	**0.02**	39 (41.5)	23 (50.0)	0.54
31–60	38 (22.6)	8 (22.2)	10 (25.6)		6 (12.2)	30 (28.8)		23 (24.5)	8 (17.4)	
≥61	52 (31.0)	9 (25.0)	11 (28.2)		22 (44.9)	26 (25.0)		32 (34.0)	15 (32.6)	
**GMFCS level, *n* = 182**										
I–II	82 (45.1)	22 (57.9)	13 (28.9)	**0.01**	37 (67.3)	37 (33.9)	**<0.001**	54 (52.4)	15 (31.3)	**0.01**
III–V	100 (54.9)	16 (42.1)	32 (71.1)		18 (32.7)	72 (66.1)		49 (47.6)	33 (68.8)	
**MACS level, *n* = 177**										
I–II	88 (49.7)	17 (47.2)	16 (37.2)	0.14	35 (66.0)	45 (42.5)	**0.002**	58 (58.6)	17 (35.4)	**0.02**
III–V	72 (40.7)	10 (27.8)	21 (48.8)		12 (22.6)	55 (51.9)		32 (32.3)	26 (54.2)	
Not applicable ^7^	17 (9.6)	9 (25.0)	6 (14.0)		6 (11.3)	6 (5.7)		9 (9.1)	5 (10.4)	
**Viking speech level ^6^, *n* = 180**										
I–II	61 (33.9)	12 (31.6)	11 (24.4)	0.25	25 (46.3)	32 (29.6)	**0.02**	40 (39.6)	14 (29.2)	0.46
III–IV	102 (56.7)	17 (44.7)	28 (62.2)		23 (42.6)	70 (64.8)		52 (51.5)	29 (60.4)	
Not applicable	17 (9.4)	9 (23.7)	6 (13.3)		6 (11.1)	6 (5.6)		9 (8.9)	5 (10.4)	
**CFCS level ^6^, *n* = 180**										
I–II	82 (45.6)	20 (52.6)	15 (33.3)	0.08	32 (59.3)	43 (39.8)	**0.02**	55 (54.5)	17 (35.4)	**0.03**
III–V	98 (54.4)	18 (47.4)	30 (66.7)		22 (40.7)	65 (60.2)		46 (45.5)	31 (64.6)	

^1^ WAZ was calculated for children aged 0–121 months and the overweight children (WAZ > +2SD) were excluded from analysis because of small number (*n* = 4); ^2^ HAZ > +2 SD were excluded from analysis due to very small counts (*n* = 6); ^3^ BAZ > +2 SD were excluded from analysis due to very small counts (*n* = 13); ^4^ Chi-square test (two-tailed); ^5^ Fisher’s exact test (two tailed); ^6^ Missing data; ^7^ MACS was assessed for children at or over four years of age. The *p* values in ‘**bold**’ font indicate significant difference.

**Table 4 nutrients-13-02537-t004:** Predictors of underweight, stunting and thinness among children with CP in Gorkha, Nepal (Adjusted analysis).

Predictors ^1^	Underweight (WAZ < −2 SD), *n* = 45		Stunting (HAZ < −2 SD), *n* = 109		Thin (BAZ < −2 SD), *n* = 48	
	aOR [95% CI]	*p* Value	aOR [95% CI]	*p* Value	aOR [95% CI]	*p* Value
**Sex**						
Male	*Ref*		*Ref*		-	
Female	2.1 (0.5, 9.2)	0.30	1.8 (0.6, 5.6)	0.28		
**Attendance to mainstream school**						
Yes	*Ref*		*Ref*		*Ref*	
No	3.8 (0.9, 16.8)	0.08	4.0 (1.2, 13.2)	**0.02**	2.2 (0.7, 6.3)	0.15
**GMFCS level**						
I–II	*Ref*		Ref		*Ref*	
III–V	2.9 (0.6, 13.0)	0.16	8.2 (1.6, 40.8)	**0.01**	1.6 (0.6, 4.1)	0.31
**MACS level**						
I–II	-		*Ref*		-	
III–V			0.1 (0.0, 0.9)	**0.04**		
**CFCS level**						
I–II	-		*Ref*		*Ref*	
III–V			1.7 (0.4, 7.1)	0.44	1.3 (0.5, 3.2)	0.58
**Viking speech level**						
I–II	-		*Ref*		-	
III–V			1.2 (0.3, 4.3)	0.77		
**Intellectual impairment**						
No	-		*Ref*		-	
Yes			2.5 (0.8, 8.2)	0.11		

^1^ Predictors that were found significantly associated in unadjusted logistic model, i.e., Appendix A, were included in adjusted model. ‘*Ref*’ indicates reference category.

## Data Availability

The data presented in this study are available on request from the corresponding author. The data are not publicly available due to privacy/ethical restrictions.

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
