# Peer review of "Nutritional Status of Children with Cerebral Palsy in Gorkha, Nepal: Findings from the Nepal Cerebral Palsy Register"

_nutrients, 2021, doi:10.3390/nu13082537_

Round 1
Reviewer 1 Report
The manuscript describes identifying children with CP in Nepal and evaluating their nutritional status. Comparison was made to WHO standards.
Overall, the manuscript is clear, appropriate methodology has been used, the text is easy to follow and data presented has not been published previously. My concern is if this particular topic is of interest to the readers of this journal. I think there are more specialized journals that may be more appropriate.
Although the manuscript is fairly well written, there are numerous small mistakes in the English. It needs to be edited by a native English speaker. I have marked just a few changes that should be made.
Also, the use of first person ("we") should be avoided in scientific writing. It appears throughout the manuscript and should be revised.
Line 57 and line 60 – Writing style: the word “vulnerable” is used twice. Please use a synonym.
Line 60 such gap- maybe “this gap”
Line 64 “serve as baseline data”
Line 65, please remove “we” and remove first person throughout.
Lines 80 & 81 the word assessment is used twice
Including a pediatrician
Line 91 “and a data collection…”
Line 95 “were used to assess and define”
Line 96 The Viking Speech Scale …
Line 104 “measured in kilograms” “using a digital “
Lines 111 and 113, please remove “we” and rewrite.
Centimeters not centimeter.
There are numerous similar errors throughout. Please correct.
Author Response
Response to Reviewer 1 Comments
We would like to thank the respected reviewer for the constructive suggestions and helpful comments on our manuscript titled “Nutritional status of children with Cerebral Palsy in Gorkha, Nepal: Findings from the Nepal Cerebral Palsy Register”.
Please see below our point-by-point response to the comments.
Point 1: The manuscript describes identifying children with CP in Nepal and evaluating their nutritional status. Comparison was made to WHO standards.
Overall, the manuscript is clear, appropriate methodology has been used, the text is easy to follow and data presented has not been published previously. My concern is if this particular topic is of interest to the readers of this journal. I think there are more specialized journals that may be more appropriate.
Response 1: Thank you for the valuable comments and suggestions.
Point 2: Although the manuscript is fairly well written, there are numerous small mistakes in the English. It needs to be edited by a native English speaker. I have marked just a few changes that should be made.
Response 2: Thank you for the suggestion and the corrections. We have read the manuscript carefully for any grammatical errors. Additionally, a native English speaker has reviewed and edited the manuscript as suggested. Please see the acknowledgement section of the revised manuscript for detail information (line 420-422).
Point 3: Also, the use of first person ("we") should be avoided in scientific writing. It appears throughout the manuscript and should be revised.
Response 3: Thank you for the valuable suggestion. We have now edited the manuscript accordingly.
Point 4: Line 57 and line 60 – Writing style: the word “vulnerable” is used twice. Please use a synonym.
Response 4: Thank you for the correction. We have edited the sentence as suggested. Please see line 59 and 62.
Point 5: Line 60 such gap- maybe “this gap”
Response 5: Thank you for the correction. We have made the changes as suggested. Please line 62.
Point 6: Line 64 “serve as baseline data”
Response 6: Thank you for the correction. We have made the necessary changes. Please see the line 66.
Point 7: Line 65, please remove “we” and remove first person throughout.
Response 7: Thank you. We have made the necessary changes as suggested.
Point 8: Lines 80 & 81 the word assessment is used twice
Response 8: Thank you for the correction. We have now edited the sentence accordingly. Please see the line 85 and 86.
Point 9: Including a pediatrician
Response 9: Thank you for the correction. We have made the changes accordingly. Please see the line 86 and 87.
Point 10: Line 91 “and a data collection…”
Response 10: Thank you for the correction. We have edited the statement accordingly. Please see the line 97.
Point 11: Line 95 “were used to assess and define”
Response 11: Thank you for the correction. We have edited the sentence accordingly. Please see the line 101.
Point 12: Line 96 The Viking Speech Scale …
Response 12: Thank you for the correction. The sentence has been corrected. Please see the line 102.
Point 13: Line 104 “measured in kilograms” “using a digital “
Response 13: Thank you for the correction. We have made the corrections accordingly. Please see the line 111.
Point 14: Lines 111 and 113, please remove “we” and rewrite.
Response 14: Thank you for the suggestion. We have made the necessary changes accordingly.
Point 15: Centimeters not centimeter.
Response 15: Thank you. We have made the correction as suggested. Please see the line 117.
Point 16: There are numerous similar errors throughout. Please correct.
Response 16: Thank you for the keen observations and the corrections. We have now read the manuscript carefully and made necessary changes as suggested.
Yours sincerely,
Israt Jahan
On behalf of the study investigators
Reviewer 2 Report
I have read the manuscrip in proof and I think it is very interesting. I have some suggestions for the authors.
- ABSTRACT. There is a sentence in the abstract (also present in the 1st paragraph of the discussion) that says "Wasting was significantly higher among children with CP aged<5years than the national average (p=0.03)". But I have not found in the text data about wasting prevalence in the region, and reference #30 is about a study from Ghana, not from Nepal. The p value here does not provide relevant information. Authors must include the data of wasting in general population with a proper reference.
- MATERIAL AND METHODS. In the 1st paragraph I miss data about the total population of the area.
- RESULTS: authors should about to repeat in the text p values that are already present in tables. Instead, the might provide more informative values (proportions, confidence intervals, etc.) when appropiate.
- RESULTS/3.2. Socio-demographic characteristics. Authors should clarify better which difference are statistically significant. For example, "underweight, stunting and thinness were more prevalent among children with low monthly family income compared to others in the cohort (p=0.03, p=0.09 and p=0.93 respectively)" would be better expressed as "underweight were more prevalent among children with low monthly family income compared to others in the cohort (59.1% if <150 USD vs 13.5% if >300 USD)".
- RESULTS/3.3.3. Predominant type, motor function severity. Again authors mergue significant results with other that are not statiscally significant, as in "Underweight, stunting and thinness were more prevalent among children with spastic CP (p=0.02, p<0.001 and p=0.45 respectively,...), tri/quadriplegia (p=0.54, p=0.001, p=0.18 respectively)". I suggest "Underweight and stunting were more prevalent among children with spastic CP, stunting among those with tri/quadriplegia,..."
- Table S2. There is no mention here (nor in Material and Methods) to what statistic is used for the correlation (¿Person's coefficient?).
- RESULTS/3.3.4. Associated impairments. Again there is a mix of significant and non-significant results in the same sentece: "Furthermore, both underweight and stunting were more prevalent among children with intellectual impairment (p=0.06 and p=0.01 respectively)". Authors may clarify this ("Underweight, but not stunting, was more prevalent among children with intellectual impairment") or simply include mention only to significant result ("Underweight was more prevalent among children with intellectual impairment").
Author Response
Response to Reviewer 2 Comments
We would like to thank the respected reviewer for the constructive suggestions and helpful comments on our manuscript titled “Nutritional status of children with Cerebral Palsy in Gorkha, Nepal: Findings from the Nepal Cerebral Palsy Register”.
Please see below our point-by-point response to the comments.
Point 1: I have read the manuscript in proof and I think it is very interesting. I have some suggestions for the authors.
Response 1: Thank you for the valuable comments and suggestions. We have incorporated all suggestions which has substantially improved the quality of the manuscript. Please see our responses to the subsequent comments.
Point 2: ABSTRACT. There is a sentence in the abstract (also present in the 1st paragraph of the discussion) that says "Wasting was significantly higher among children with CP aged<5years than the national average (p=0.03)". But I have not found in the text data about wasting prevalence in the region, and reference #30 is about a study from Ghana, not from Nepal. The p value here does not provide relevant information. Authors must include the data of wasting in general population with a proper reference.
Response 2: Thank you for the correction. We sincerely apologize for the mistake. There was a mix-up in the bibliography which has been corrected now in the revised manuscript. Please see the edited reference list (line 498-503). We have also reviewed the manuscript carefully to remove any error in the referencing. Furthermore, we have edited the statement in the abstract accordingly. Please see line 29.
Point 3: MATERIAL AND METHODS. In the 1st paragraph I miss data about the total population of the area.
Response 3: Thank you for the comment. We have added this information in the edited manuscript. Please see line 76-77.
Point 4: RESULTS: authors should about repeat in the text p values that are already present in tables. Instead, they might provide more informative values (proportions, confidence intervals, etc.) when appropriate.
Response 4: Thank you for the valuable suggestion. We have now edited the results section accordingly. Please see the results section of the revised manuscript (line 183-185, 220-239).
Point 5: RESULTS/3.2. Socio-demographic characteristics. Authors should clarify better which difference are statistically significant. For example, "underweight, stunting and thinness were more prevalent among children with low monthly family income compared to others in the cohort (p=0.03, p=0.09 and p=0.93 respectively)" would be better expressed as "underweight were more prevalent among children with low monthly family income compared to others in the cohort (59.1% if <150 USD vs 13.5% if >300 USD)".
Response 5: Thank you for the thoughtful suggestion and the corrections. We have revised the sentences accordingly. Please see line 183-185.
Point 6: RESULTS/3.3.3. Predominant type, motor function severity. Again, authors merge significant results with other that are not statistically significant, as in "Underweight, stunting and thinness were more prevalent among children with spastic CP (p=0.02, p<0.001 and p=0.45 respectively,), tri/quadriplegia (p=0.54, p=0.001, p=0.18 respectively)". I suggest "Underweight and stunting were more prevalent among children with spastic CP, stunting among those with tri/quadriplegia,"
Response 6: Thank you for the valuable suggestion. We have edited the sentences accordingly. Please see line 219-227.
Point 7: Table S2. There is no mention here (nor in Material and Methods) to what statistic is used for the correlation (¿Person's coefficient?).
Response 7: Thank you for the keen observation. We sincerely apologize for not clarifying it earlier. We previously used the Pearson’s coefficient; however, we have now deleted the S2 from the results section. We have also removed the relevant information from the text in the results section.
Point 8: RESULTS/3.3.4. Associated impairments. Again, there is a mix of significant and non-significant results in the same sentence: "Furthermore, both underweight and stunting were more prevalent among children with intellectual impairment (p=0.06 and p=0.01 respectively)". Authors may clarify this ("Underweight, but not stunting, was more prevalent among children with intellectual impairment") or simply include mention only to significant result ("Underweight was more prevalent among children with intellectual impairment").
Response 8: Thank you for the valuable suggestion. We have edited the sentences accordingly. Please see the line 237-238 of the revised manuscript.
Yours sincerely,
Israt Jahan
On behalf of the study investigators